# Stochastic Ultralow-Frequency Oscillations of the Luminescence Intensity from the Surface of a Polymer Membrane Swelling in Aqueous Salt Solutions

**DOI:** 10.3390/polym14040688

**Published:** 2022-02-11

**Authors:** Nikolai F. Bunkin, Polina N. Bolotskova, Elena V. Bondarchuk, Valery G. Gryaznov, Valeriy A. Kozlov, Maria A. Okuneva, Oleg V. Ovchinnikov, Oleg P. Smoliy, Igor F. Turkanov, Catherine A. Galkina, Alexandr S. Dmitriev, Alexandr F. Seliverstov

**Affiliations:** 1Department of Fundamental Sciences, Bauman Moscow State Technical University, 2nd Baumanskaya Str. 5, 105005 Moscow, Russia; bolotskova@inbox.ru (P.N.B.); v.kozlov@hotmail.com (V.A.K.); neonlight0097@gmail.com (M.A.O.); 2Prokhorov General Physics Institute, Russian Academy of Sciences, Vavilova Str. 38, 119991 Moscow, Russia; 3“Concern GRANIT”, Gogolevsky Blvd., 31, 2, 119019 Moscow, Russia; info@npo-qt.ru (E.V.B.); gryaznov.v@granit-concern.ru (V.G.G.); office@granit-concern.ru (O.V.O.); smoliy.o@granit-concern.ru (O.P.S.); turkanov.i@granit-concern.ru (I.F.T.); galkina.e@granit-concern.ru (C.A.G.); 4Kotelnikov Institute of Radio Engineering and Electronics of the Russian Academy of Sciences, Mokhovaya 11, 7, 125009 Moscow, Russia; alexandrdm@bk.ru; 5Frumkin Institute of Physical Chemistry and Electrochemistry of the Russian Academy of Sciences, Leninsky Prospect 31, 4, 119071 Moscow, Russia; alex_sel@bk.ru

**Keywords:** photoluminescence spectroscopy, isotonic solution, low frequency electromagnetic radiation, bubston clusters, resonant energy transfer of the luminescent state

## Abstract

Photoluminescence from the surface of a Nafion polymer membrane upon swelling in isotonic aqueous solutions and Milli-Q water has been studied. Liquid samples were preliminarily processed by electric pulses with a duration of 1 μs and an amplitude of 0.1 V using an antenna in the form of a flat capacitor; experiments on photoluminescent spectroscopy were carried out 20 min after this treatment. A typical dependence of the luminescence intensity, *I*, on the swelling time, *t*, obeys an exponentially decaying function. The characteristic decay time of these functions and the stationary level of luminescence intensity depend on the repetition rate of electrical pulses, and the obtained dependences are well reproduced. It transpired that, at certain pulse repetition rates, the dependence, *I*(*t*), is a random function, and there is no reproducibility. Stochastic effects are associated with a random external force of an electromagnetic nature that acts on a polymer membrane during swelling. The source of this random force, in our opinion, is low-frequency pulsations of neutron stars or white dwarfs.

## 1. Introduction

Nafion^TM^ (C_7_HF_13_O_5_S × C_2_F_4_) consists of perfluoro-vinyl ether groups terminated with sulfonic groups on a tetrafluoroethylene (Teflon) backbone. Teflon is very hydrophobic, while the sulfonic groups are essentially hydrophilic. While swelling in an aqueous media, a nanostructure consisting of cylindrical reverse micelles appears. Water-filled channels of 2–3 nm diameter form within the Nafion membrane; see [1] for more details. The polymer Nafion is widely studied in various fields, such as physics, chemistry, and hydrogen energetics (see, e.g., references [2,3,4,5,6], related to the articles issued in 2021). The majority of techniques, applied for Nafion studies, are focused on the study of the polymer bulk properties. At the same time, water adjacent to the swollen polymer surface has also been explored, see recent works [7,8,9,10], and the monograph [11] describes experiments, in which a Nafion membrane is immersed in an aqueous suspension of colloidal microspheres. It transpired that the microspheres are repelled from the membrane up to a distance of several hundreds of microns. The area, from which the colloidal microspheres are effectively pushed out, has been termed the “exclusion zone” (EZ). EZ phenomena may have important engineering applications in water filtration, reducing biofouling [12] and microfluidics [13]. EZ phenomena also have an obvious importance to understanding biological systems and resolving outstanding questions regarding “biological water” [14]. In accordance with the model developed in monograph [11], Nafion’s surface imparts a quasicrystalline structure on a macroscopic scale to adjacent water layers; in monograph [11] (see also numerous references therein), this effect was called the formation of the “fourth phase” of water.

The width of the exclusion zone is approximately 300 µm. This is astonishing and not explicable by standard physical chemistry theories. Some physical mechanisms that might be responsible for the EZ formation have been discussed in a recent review [15]. As was shown in our recent work [16], when Nafion swells in water, polymer fibers unwind into the water bulk, but do not completely detach from the membrane surface, thus forming a stiff brush-like structure. Thus, according to our model, the EZ is not a special phase of water, and colloidal microspheres are pushed out of the area adjacent to the membrane due to the unwinding of the polymer fibers. It is very important that the effect of unwinding depends on the deuterium content in a liquid sample; for more detail see [16]. In fact, for ordinary natural water (the deuterium content is 157 ± 1 ppm, see [17]), the EZ size amounts to several hundreds of microns due to unwinding, while for the so-termed deuterium depleted water (DDW, the deuterium content is ≤1 ppm), the effect of unwinding is missing, and the EZ size is zero. Bearing in mind that the Nafion membrane is “decorated” with unwound polymer fibers, we see a clear analogy with human cell membrane. Indeed, the structure of the channels in the bulk of the polymer is similar to the lipid bilayer, while the unwound fibers are similar to the glycocalyx (extracellular matrix) and endothelial surface layer of vascular tissue of humans; see, for example, [18]. This is a matter of utmost importance to physiology; see [19]. Developing the analogy between the human cell membrane and the Nafion membrane, we note that there are a number of organic compounds that stimulate the permeability of various ions through the cation channels of the cell membrane. We can say that the Nafion polymer membrane is similar to synthetic biological polymer membranes; see the recent review [20]. In this regard, the alkaloid piperine (C_17_H_19_NO_3_), which stimulates the conductivity of calcium [21] and potassium [22] ions in human cell channels, is of particular interest to study. Indeed, because the Nafion membrane contains negatively charged channels that are the conductors of protons, it is very important to study the interaction of piperine with Nafion.

One of the goals of this work is to clarify the mechanisms of interaction of cell membranes with non-ionizing low-frequency electromagnetic radiation. This interaction underlies the treatment of various diseases with a low-frequency electromagnetic field; see [23,24,25,26,27,28,29,30]. The cited monographs and reviews confirm the effectiveness of this technique. These works were written mainly by physicians who use this technique in their medical practice; the physical nature of the low-frequency effect on living tissue is usually not discussed. Within the framework of the analogy between the cell membrane and the Nafion membrane, it is of interest to study the features of the interaction of Nafion with low-frequency electromagnetic radiation.

We, in our group, performed a number of experiments on the effect of low-frequency electromagnetic irradiation of isotonic solutions, in which the Nafion membrane was then soaked; see [31]. As shown in [31], the characteristic time of swelling of polymer membrane is different for irradiated/non-irradiated (reference) liquid samples. The time interval between electromagnetic treatment and measurements of the irradiation effects was about 20 min. The purpose of this study is to clarify the mechanisms of interaction of a polymer membrane with isotonic solutions with low-frequency electromagnetic radiation, which can be used in medical practice. In this work, we performed a number of experiments with photo-luminescent spectroscopy, where we studied in detail the range of repetition rates of electrical pulses from 5 Hz to 500 kHz with an interval of 5 Hz; if at certain frequencies any features were found in the dynamics of polymer swelling (see below), the corresponding frequency interval was passed with a step of 1 Hz. In some experiments we covered our experimental setup with aluminum foil. Furthermore, we investigated liquid samples with different isotopic compositions. This allowed us to detect a number of isolated frequencies of electromagnetic processing, at which the luminescence intensity, usually described by a regular function of time, exhibits a sporadic temporal behavior. The novelty of our work consists of the study of the spectral characteristics of the stochastic dynamics of luminescence and a qualitative theoretical model underlying the stochastic regimes, explaining the differences in the swelling kinetics of polymer in natural water and DDW. The results can be used to clarify the mechanisms of interaction of the polymer membrane with isotonic solutions with different isotopic compositions in the field of external low-frequency irradiation, which can be applied in medical practice.

## 2. Materials and Methods

### 2.1. Materials

We investigated Nafion N117 plates (Sigma-Aldrich, St. Louis, MO, USA) with a 175 μm-thickness and a 1 × 1 cm^2^ square area. The Nafion plates were soaked in Milli-Q water with a resistivity of 4 MΩ × cm (measurement were made 1 h after the preparation with a conductometer CON270043S Eutech, Thermo Fisher Scientific, Waltham, MA, USA), as well as in isotonic NaCl (0.9%; Mosfarm, Moscow region, Russia) and Ringer’s (Biosintez, Penza, Russia) solutions. In our case, the Ringer’s solution was composed of NaCl (8.6 g/L), KCl (0.3 g/L), and CaCl_2_ × 6H_2_O (0.25 g/L), dissolved in water. The deuterium content of these samples was 157 ± 1 ppm. In some cases, reagent-grade NaCl (Sigma-Aldrich, St. Louis, MO, USA) was used to prepare 0.9% NaCl solutions based on deuterium depleted water (DDW; deuterium content ≤ 1 ppm, purchased from Sigma-Aldrich, St. Louis, MO, USA). Piperine (C_17_H_19_NO_3_), 98%, was purchased from CheMondis GmbH, Cologne, Germany. Mixtures of piperine were prepared in Ringer’s solution with a concentration of 40 mg/L; this concentration corresponds to a saturated mixture.

### 2.2. Instrumentation

#### 2.2.1. Processing of Liquid Samples

To study the effects of electromagnetic treatment, we designed an experimental setup that allowed us to expose liquid samples to electrical pulses of positive/negative polarity with 5 Hz–500 kHz repetition rate with the possibility of varying the amplitude of pulses; the pulse duration was 1 μs. The photo of this setup is given in Figure 1a,b.

In these experiments, we used a G5-63 generator of pulses (Priborelektro, Moscow, Russia), and an oscilloscope AKIP 4115/3A (Novapribor, Moscow, Russia). This setup allowed us to irradiate the liquid samples in a contactless mode; see Figure 1a. A glass cell of 200 mL contained the liquid samples under study. This cell was placed inside a planar capacitor; the area of electrodes was 100 cm^2^, and the distance between them was 5 cm, i.e., the capacitance was 1.77 pF. The results reported below are related to 100 mV amplitude of a pulse; see Figure 1b. Liquid samples were exposed to electric pulses for 20 min (the time of processing). After irradiation of a liquid sample, the processed liquid was poured into the cell shown in Figure 2 to study the photoluminescence signal from the Nafion plate. The interval between the end of electric pulses processing and the beginning of the luminescence experiment was ~20 min. It is generally accepted that the characteristic relaxation times in water and aqueous solutions are limited by the hydrogen bond lifetime, being approximately a picosecond; see, e.g., [32]. Thus, it would seem likely that all the effects associated with electromagnetic treatment should not reveal 20 min after this treatment.

#### 2.2.2. Photoluminescence Study

In this subsection we briefly describe the protocol for the photoluminescence experiments; for more detail, see [16,31]. The technique is based on the luminescence excitation from the Nafion surface due to the pumping in the UV range. To excite luminescence, it is necessary to irradiate a substance within one of its absorption bands. It is known [33] that the absorptivity maxima of Nafion is centered at a wavelength of λ = 270 nm. It is also known (see, for example, [34]) that water does not absorb in this spectral range. In our photoluminescence experiments we used radiation at a λ = 369 nm wavelength corresponding to the long-wavelength region of the absorption band mentioned. In parallel experiments (see [16]), it was found that the terminal sulfonic groups, HSO_3_, serve as the centers of Nafion luminescence upon UV irradiation.

As shown in the experiments with the Nafion solution in isopropanol, (see [16]), the luminescence intensity, *I*, grows linearly with an increasing of the volume number density of Nafion particles, *n_Naf_*, and can be expressed as:(1)$$I=A+kI_{pump}\sigma_{lum}n_{Naf}V,$$
of the luminescence centers, i.e., terminal sulfonic groups HSO_3_. Because these groups are attached to polymeric chains, *n_Naf_* can be associated with the volume number density of Nafion particles. Here *I_pump_* is the pump intensity, *A* = 20–270 arbitrary units correspond to the spectral density of the mini-spectrometer noise and stray-light illumination in relative units, *k* is the setup transfer coefficient, *V* is the luminescence volume, and *σ_lum_* is the luminescence cross section (the spectral maximum of *σ_lum_* corresponds to λ = 460 nm). The linear dependence of the luminescence intensity, *I* vs. *n_Naf_*, is realized, providing that *σ_lum_* = const.

To carry out experiments on the photoluminescence spectroscopy, a setup was designed (for more detail, see [13]). The schematic of the setup is shown in Figure 2.

The probing radiation of the continuous wave laser diode (1) (optical pumping) at a wavelength of *λ* = 369 nm was introduced into the multimode Ø = 100 μm-diameter quartz optical fiber (2) with a numerical aperture, *NA* = *n*·sinα = 0.3, where *n* = 1 is the air refractive index and α is the beam divergence angle at the exit end of the fiber in air. The fiber was fixed in a hole at the center of the bottom of a cylindrical cell (3) made of Teflon; the direction of pump beam set the experimental setup optical axis. The cell was thermostabilized at room temperature (*T* = 23 °C), accurate to ±0.1 °C, and filled with the test liquid sample. We studied the swelling of a square Nafion plate (4) of a side *h* = 10 mm and a thickness *d* = 175 μm. The plate was fixed in parallel to the optical axis, i.e., the experiments were carried out at grazing incidence geometry. The Nafion plate edges were tightly fixed with two clamps; no additional means for the plate fixation were used. The size of the clamps was much less than the Nafion plate width, i.e., the approximation of free boundary conditions was realized. Additionally, the thickness of the Nafion plate could increase in the process of swelling; this effect was minimized due to the rigid fixation of the plate.

At the beginning of the experiment, a dry (water-free) Nafion plate was placed in an empty cell; we could move the plate horizontally using a stepping motor (8), thereby changing the plate position with respect to the optical axis. Doing so, we achieved the maximum of the luminescence signal; this position of the Nafion plate was considered to be optimal. When a liquid sample was poured into the cell, the initially hydrophobic Nafion plate was bent along the optical axis. However, such bending led only to an effective shift of the Nafion-water boundary (this shift amounted to approximately 1 mm) but did not result in a change in the pump radiation incident angle. To restore the optimal plate position with respect to the optical axis, additional adjustment was carried out using a stepping motor (8). The luminescence radiation was reflected by the cell internal surface (Nafion is transparent in the visible range) and collected along the cell optical axis. This resulted in a significant gain of luminescence intensity. The luminescence signal was received by a quartz fiber (5), fixed at the center of the cell, and transferred to the minispectrometer (6) FSD-8 (Russia). The experimental data were accumulated by a computer (7). In the experiment, the temporal dynamics of the luminescence intensity in its spectral maximum (λ = 460 nm) depending on the time of Nafion soaking in the liquid under study was investigated. The start of counting the soaking time, *t*, corresponds to the moment of pouring the liquid sample into the cell.

## 3. Experimental Results

Figure 3a–c shows the typical dependences of *I* at its spectral maximum (λ = 460 nm) vs. swelling time, *t*, in Milli-Q water, isotonic NaCl, and Ringer solutions; for definiteness, here we present the dependences for 60 Hz pulse repetition rate. Reference curves for untreated (reference) samples are also shown. The experimental points correspond to averaging over five successive measurements. Confidence intervals are indicated in all graphs. All obtained dependences are well approximated by a single exponential function with characteristic decay times, *τ*; the corresponding formulas are given in the insets and highlighted with different colors. As follows from the graphs, the reference dependences are described by approximately the same exponents: Y = 14,366 + 47,655⋅exp(−*t*/14) for water, Y = 13,239 + 45,548⋅exp(−*t*/14) for NaCl solution, and Y = 18,937 + 48,400⋅exp(−*t*/13) for Ringer’s solution, i.e., the ionic additives would hardly influence the Nafion swelling dynamic. In this particular case, the luminescence intensity in all treated samples is lower than the intensity in the reference samples; note that for some pulse repetition rates the situation changes, i.e., the luminescence intensity in the treated samples can exceed the reference dependence or coincide with the reference curve to a good accuracy. The main feature of these results are as follows: the luminescence intensity *I*(*t*) after treatment is described by a single exponential function, and the results obtained have fairly good reproducibility. Similar results (at other frequencies) were presented in [31], so this will not be discussed in detail here.

The exponential dependence of *I*(*t*) means that the luminescence cross section, *σ_lum_*, is constant. Indeed, assuming that the volume number density, *n_Naf_*, of the luminescence centers (sulfonic groups in our case) obeyed Equation (1), i.e., *n_Naf_* in the near-surface polymer layer decreases due to penetrating of water molecules into this layer, we can write:(2)$$\frac{dn_{Naf}}{dt}=-\frac{n_{Naf}}{\tau },$$
where *τ* is the characteristic time of swelling. Thus, we obtain:(3)$$n_{Naf}={\left(n_{Naf}\right)}_{0}\exp\left(-\frac{t}{\tau }\right).$$

Because the *I*(*t*) is proportional to *n_Naf_* (see Equation (1)), it should exponentially decay.

We see that the processing of liquid samples with electrical pulses does not lead to significant changes in the kinetics of the swelling of the polymer; the exponential behavior of *I*(*t*) remains after treatment. It transpired, however, that the addition of some organic compounds to the liquid samples under study can lead to a violation of the exponential behavior of *I*(*t*), and this does not require treatment with electric pulses. In Figure 4 we exhibit the dependence of *I*(*t*) for the mixture “Ringer’s solution-piperine” with a piperine concentration of 40 mg/L (saturated mixture). We see that, in this case, the dependence of *I*(*t*) is rather random; the results of more detailed studies of mixtures of piperine in water and isotonic solutions, including the treatment of liquid samples with electric pulses, will be published elsewhere. It is important for us that a stochastic character in the behavior of *I*(*t*) can arise, for example, at the adding of certain organic compounds to the test liquids.

We will further consider only NaCl isotonic solutions. In Figure 5 we show the luminescence intensity, *I*(*t*), dependences at the spectral maximum for 400 and 440 Hz pulse repetition rates. In contrast to the situation depicted in Figure 4, the graphs in Figure 5 are quite close to the reference curves and are described by decaying exponential functions, which are well reproduced.

However, there exist a number of pulse repetition frequencies for which the treatment by electrical pulses leads to irregular temporal behavior of *I*(*t*), that is, the effect of treatment is similar to the addition of piperine to a liquid sample. We found the following peculiar isolated frequencies: 82.5, 174, and 417 Hz. The values of these frequencies obey the approximate ratio 1:2:5. Perhaps there exist some other special frequencies, but we have not found them yet. We also do not know the physical mechanism behind the effects, connected with these frequencies. Hereinafter, we will, for definiteness, restrict ourselves with the case of 417 Hz repetition frequency and isotonic NaCl solution. In Figure 6a,b (cf. graphs in Figure 5 for 400 and 440 Hz frequencies) we demonstrate the dependencies in several successive experiments performed two weeks apart in different laboratories; the distance between these laboratories is approximately 20 km. The intervals between the start of measurements for each experiment are denoted by the letter *τ*. Obviously, there is no good reproducibility from measurement to measurement, so the confidence intervals are not shown. We also give the results of calculations of Pearson rank correlation coefficient, R, between the graphs.

The results of spectral and correlation processing of the experimental curves presented in Figure 6 are shown in Figure 7, Figure 8 and Figure 9. In Figure 7, we exhibit the dependences, *A*(*f*), which is the Fourier transform of *I*(*t*) shown in Figure 6b for various *τ*; these dependences were obtained by using the Morlet wavelet transform; see [35]. In our particular case, the Morlet wavelet is given by the formula $\psi \left(t\right)=\exp\left(-t^{2}/2\right)\cos\left(5t\right).$ Here, *A*(*f*) is the spectral amplitude at frequency *f*. It is seen that there exists a spectral maximum at a frequency of ~4 × 10^−4^ Hz on all curves. It is also seen that at the frequency *f* → 0 the dependence *A*(*f*) diverges, which is related to the specifics of the wavelet transforms using the Morlet wavelet at ultralow frequencies (see [35]), i.e., the growth of the function *A*(*f*) at low frequencies can be ignored. In further analysis, a segment of *A*(*f*) with an isolated spectral maximum was allocated from the dependences *A*(*f*), and for this segment the spectral density, *A*^2^(*f*), was found, which, in turn, was approximated by Lorentzian:(4)$$L\left(f\right)=C+\frac{B}{\pi }\cdot \frac{1/\tau_{corr}}{4{\left(f-f_{0}\right)}^{2}+{\left(1/\tau_{corr}\right)}^{2}}.$$
Here *C* and *B* are the constants, f_0_ is the central (resonant) frequency, and *τ_corr_* is the correlation time of random process with the Lorentzian spectral line; the Lorentzian contour width is Δ*f*~(*τ_corr_*)^−1^, see [36]. In Figure 8, we show the dependence, *A*^2^(*f*), for the curve, *A*(*f*), shown in Figure 7 for *τ* = 52.5 h. In Figure 9a,b, we exhibit the dependences of the Pearson rank correlation coefficient, *R*, vs. time, *τ*. A detailed description of the methods for calculating the Pearson rank correlation coefficient can be found in the hyperlink [37]. In our case, Pearson’s correlation coefficient, when applied to a sample, is described by the formula:(5)$$R=\frac{\sum_{i=1}^{n}\left(x_{i}-\bar{x}\right)\left(y_{i}-\bar{y}\right)}{\sqrt{\sum_{i=1}^{n}{\left(x_{i}-\bar{x}\right)}^{2}}\sqrt{\sum_{i=1}^{n}{\left(y_{i}-\bar{y}\right)}^{2}}}.$$
where *n* is sample size and *x_i_* and *y_i_* are the individual sample points indexed with *i*, $\bar{x}=\frac{1}{n}\sum_{i=1}^{n}x_{i}$ (the sample mean; analogously for $\bar{y}$). In Formula (5), the set of values *x_i_* corresponds to the first measurement of *I*(*t*) in Figure 6 (indicated by the symbol (1) on the graphs), and the set of values *y_i_* corresponds to the measurement of *I*(*t*) after a time, *τ*. As follows from Equation (5), *R*(*τ* = 0) = 1.

In addition, in some experiments we wrapped the flask containing the liquid sample with 10 μm-thick aluminum foil, and the sample was treated with electric pulses at 417 Hz repetition frequency. The results of this experiment are shown in Figure 10. It is seen that, in this case, no stochastic behavior was observed, and the luminescence intensity, *I*(*t*), coincides with a good accuracy with the reference curve. This is quite expected: the treatment effect in this case is absent due to the screening of the liquid sample.

In addition, in some experiments we covered the photoluminescence setup (see Figure 2) with 10 μm-thick aluminum foil; in these experiments, we also studied the luminescence from Nafion, swollen in isotonic solution of NaCl treated with electric pulses at 417 Hz frequency. In Figure 11, we present two graphs obtained during the same experiment: without the foil screen and with the screen. We can see that if the setup is screened, the stochastic behavior is absent, and the luminescence intensity, *I*(*t*), is close to the reference curve. However, as the screen was removed, stochastic jumps in the luminescence signal appeared once again. This behavior can be explained by the effect of an external random force of an electromagnetic nature, which is manifested during the luminescent spectroscopy experiment.

As was mentioned in the Introduction Section, in the process of swelling, the polymer fibers are effectively unwound into the bulk of liquid, but do not completely detach from the surface, i.e., a rigid brush-like structure is formed; the length of the area, occupied with unwound fibers, amounts to hundreds of microns. As shown in [16], the unwinding effect depends on the deuterium content in a liquid sample. In fact, this effect is absent in the so-termed deuterium depleted water (DDW, the content of deuterium is ≤1 ppm). Thus, it is necessary to perform the luminescence experiment with the DDW-based liquid sample. In Figure 12, we show the luminescence intensity, *I*(*t*), in isotonic NaCl DDW-based solution. It is seen that no stochastic oscillations are observed, i.e., the luminescence intensity, *I*(*t*), coincides with a good accuracy with the reference curve.

## 4. Discussion

The stochastic behavior of the intensity, *I*(*t*), shown in the graphs of Figure 6 can be explained by the effects associated with an external low-frequency electromagnetic field. This follows from the fact that the random character of *I*(*t*) disappears when the experimental setup is screened with aluminum foil (see Figure 11), but it appears once again when the screen is removed. As follows from the graph in Figure 8, the spectral density of this electromagnetic wave is centered at the frequency *f*_0_ = 3.8 × 10^−4^ Hz, and the spectral line width at the half-height level is Δ*f* = 1.416 × 10^−4^ Hz (see the inset in Figure 8). The value of Δ*f* is consistent with the dependences of the correlation coefficient *R*(*τ*); see Figure 9a,b. As is seen in Figure 9a,b, the experimental points fit well the decaying exponential function *R*(*τ*) ~ exp(−*τ*/*τ_corr_*), where *τ_corr_* is the decay time (correlation time). As follows from the graphs in Figure 9, it does not matter from what moment of time we started the measurements of *I*(*t*). In addition, the results obtained are indifferent to the spatial position of the experimental setup. Thus, we are apparently dealing with stationary and spatially homogeneous external electromagnetic waves. As follows from the Wiener–Khinchin theorem [36], the dependence *R*(*τ*) ~ exp(−*τ*/*τ_corr_*) corresponds to the spectral density of a random process, described by Lorentzian contour of width Δ*f* ~ (*τ_corr_*)^−1^. In the case of pattern, exhibited in Figure 9a, we have *τ_corr_* = 25,740 s and Δ*f* ≈ 0.4 × 10^−4^ Hz. In the case of pattern in Figure 9b, we have *τ_corr_* = 21,780 s and Δ*f* ≈ 0.5 × 10^−4^ Hz. Thus, the values of Δf obtained on the basis of the data in the graphs of Figure 8 and Figure 9 are of the same order of magnitude. We still do not know why this stochastic behavior reveals only at certain frequencies of processing liquid samples.

In our opinion, an external source of electromagnetic radiation, due to which the effects described above arise, can be associated with the pulsations of distant space objects. In this regard, it is necessary to mention the recently published article [38] devoted to the discovery of an unknown phenomenon that generates a giant energy outburst with a regular time interval of 18.18 min. This cosmic phenomenon is unlike anything scientists have encountered before. Researchers believe that the mysterious source of flashes is a neutron star or a white dwarf with a super-powerful magnetic field. This object is possibly rotating, emitting a beam of radiation, which becomes a very bright radio source within 30–60 s. The object itself is located at a distance of about four thousand light years from Earth. As shown in [37], this object can be a magnetar with a very long period and an extremely strong magnetic field. This object is likely to effectively convert magnetic energy into radio waves. The frequencies of radio waves emitted during flashes of a minute duration lie in the range of 80–220 MHz, and this radiation is linearly polarized.

These flashes occur with a frequency of 1/18.18 min^−1^ = 9 × 10^−4^ Hz ≈ 2*f*_0_. Therefore, we cannot assert that the source of the radio waves leading to the stochastic behavior of *I*(*t*) corresponds to that described in [38]. It is very important for us that such sources really do exist. Therefore, without loss of generality, we will assume that we are dealing with pulses of a linearly polarized electromagnetic wave with a frequency of 100 MHz; the repetition rate of these pulses is *f*_0_ = 3.8 × 10^−4^ Hz.

Let us first show that there is no absorption of electromagnetic wave in water at the frequency *ω* = 100 MHz. As is known [39], the interaction of an external electromagnetic wave with medium is described using the complex permittivity *ε* = *ε*′ − *i**ε*″, where the real part, *ε*′, describes the “ability” of the medium to be polarized by the external field, and the imaginary part, *ε*″, describes the energy loss associated with absorption and conversion to heat. For the coefficients *ε*′ and *ε*″, we have:(6)$$\varepsilon^{\prime }\left(\omega \right)=\frac{\varepsilon_{S}-\varepsilon_{\infty }}{1+\left(\omega^{2}\tau^{2}\right)}+\varepsilon_{\infty },$$
(7)$$\varepsilon^{\prime \prime }\left(\omega \right)=\frac{\left(\varepsilon_{S}-\varepsilon_{\infty }\right)\left(\omega \tau \right)}{1+\left(\omega^{2}\tau^{2}\right)}.$$
where *τ* is the time of rotational diffusion of water molecules,
(8)$$\tau =\frac{4\pi \eta r^{3}}{kT},$$

*ε_S_* is static permittivity (for water *ε_S_* = 81), ε_∞_ is the dielectric permittivity in the optical range (for water ε_∞_ = 1.77), *r* is molecular radius (for water *r* = 1.38 Å), and *η* is the dynamic viscosity (for water *η* = 8.9 × 10^−4^ Pa·s). After the substitutions for water under normal conditions, we obtain *τ* ≈ 8.27 ps. This implies the estimate *ωτ* ≈ 8.3 × 10^−4^, that is, *ε*″ << *ε*′ ≈ *ε_S_*, and absorption at this frequency can be neglected.

As follows from Figure 6 and Figure 12, for the stochastic oscillations to occur, it is necessary that, during the swelling of the membrane, the polymer fibers must be unwound into the bulk of the liquid. We also recall that experiments on photoluminescent spectroscopy begin approximately 20 min after the end of the liquid treatment with electrical pulses. Below we present a qualitative model that describes the appearance of stochastic oscillations, taking into account the effects of long-term relaxation of liquid samples, as well as taking into account the unwinding of polymer fibers.

As shown in our recent work [40], if a liquid is saturated with a dissolved gas (for example, atmospheric air) and has an ionic component, then gas nanobubbles spontaneously arise in it. These nanobubbles are stabilized due to the anion’s adsorption on the inner (from the side of gas phase) surface of the nanobubbles. Such nanobubbles were termed bubstons (abbreviation of bubble, stabilized by ions). When considering the stabilization of bubstons due to ionic adsorption, it is necessary to take into account that negatively charged (due to the adsorption of anions) gas cores are always surrounded by a spherically symmetric diffusion cloud of counterions (Debye screening). When bubstons move in a viscous liquid, the peripheral layers of this cloud are effectively “washed away”, and the bubstons are negatively charged. Indeed, during electrophoresis the bubbles move to the anode, see [41,42,43]; regarding the methods for measuring the electrophoretic mobility of particles, see also [44,45,46].

Note that in aqueous salt solutions with a sufficiently high ionic concentration, individual bubstons are capable of coagulating to one another with the formation of micron-sized bubston clusters. As shown in [47], the bubstons, moving in viscous electrolytes solutions, acquire opposite signs, which leads to their coagulation due to centrally symmetric Coulomb attractive force:(9)$$F^{\prime }=\frac{Qq}{4\pi \varepsilon_{0}\varepsilon_{S}r^{2}}.$$

Here, *Q* is the bubston charge, which can be estimated based on the experiments with dynamic light scattering results and ζ-potential measurements; see [40]. We get *Q*~10^−17^ C, i.e., the bubston charge is approximately a hundred elementary charges. The other parameters in Equation (9) are the following: *r* is the bubston cluster radius (~1 μm) and *q* is the effective charge of the center, which attracts a bubston in the coagulation process. Thus, we obtain an estimate *F′*
$\sim 10^{-14}N$. Here we do not know the exact value of the charge, *q*; in Equation (9) we assume *q*~−*Q*. However, because the cluster is an aggregate of dimers consisting of particles, having the opposite sign, the charge, *q*, should be calculated as a sum of the terms of the alternating series. Thus, most likely, |*q*| *<< |Q|*, i.e., *F′* << 10^−14^ N. It is clear that, due to the interaction (9) a spherically symmetric bubston cluster should form. Experimental studies of bubston clusters in aqueous solutions of NaCl are presented in [48]; transpired that the cluster phase is manifested in experiments at ion concentrations >0.1 M. It is obvious that isotonic NaCl and Ringer solutions meet this condition, while in deionized water the concentration of ions is less, and the cluster phase is absent. Indeed, in accordance with our measurements, the pH value in deionized water is 5.7, i.e., the ion content is 10^−6^ M, which is not sufficient for the coagulation of bubstons, and the clusters are not manifested in experiments with dynamic light scattering in water.

The characteristic lifetime of the bubstons/bubston clusters was measured in [40]. In this experiment, the distribution of sizes of the scatterers in 1 M NaCl solution was measured, and then the liquid sample was settled for 6 months under stationary conditions in hermetically sealed (without access of atmospheric air) cell of 2-cm height. It transpired that the micron-sized scatterers (bubston clusters) disappear after this long-term settling, while submicron-sized scatterers (individual bubstons) remain in the liquid (“survive”). The disappearance of large scatterers is obviously due to their floating up, following by destroying at the liquid interface, while individual bubstons have a neutral buoyancy, i.e., they do not float up. Thus, we can argue that bubstons are an equilibrium phase of an aqueous electrolyte solution under normal conditions, and the bubston clusters are a long-lived phase.

Let us now assume that the uniform electric field of a flat capacitor, used for the treatment of the liquid samples, has a strength of *E* ~ 1 V/m (the voltage was ~0.1 V). Thus, a force *F = QE*~10^−17^ N arises in a flat capacitor, which is superimposed on the spherically symmetric Coulomb force (9). Assuming *F′*~10^−14^ N, we have an *F/F′*~10^−3^, but this is apparently underestimated, see above. Assuming *F′* ≈ *F*, we find that the force, *F*, generated inside a plane capacitor, can violate the bubston cluster’s spherical symmetry, and the cluster becomes slightly anisotropic. This was indirectly confirmed in the experiment with the NaCl solution treatment at a frequency of 417 Hz; in this case, one flat capacitor (see Figure 13a) or two flat capacitors, installed normally with each other (Figure 13b), were used. The dependence, *I*(*t*), is shown in Figure 13c. As follows from the graphs in Figure 13c, when using one flat capacitor, the luminescence intensity, *I*(*t*), behaves stochastically (red curve), while processing with two mutually perpendicular capacitors (blue curve), the run of *I*(*t*) is close to the reference dependence.

Obviously, when we are dealing with two perpendicular capacitors, the external electric field does not have a fixed axis, i.e., this field does not change the bubston cluster sphericity. In this case, the luminescence intensity is close to the reference dependence, which is fairly well reproduced. If the processing takes place in the field of one flat capacitor, then the behavior of *I*(*t*) is random. Thus, for the memory effect it is necessary to change the cluster geometry: the cluster must be slightly anisotropic, and these anisotropic properties should persist for a rather long time. Indeed, any physical mechanisms for the relaxation of an anisotropic cluster to its spherical symmetry are evidently absent.

Let us imagine that external linearly polarized radiation at a frequency of 100 MHz is scattered by bubston clusters. If the clusters are spherically symmetric, then the scattered radiation will remain linearly polarized. However, if the clusters are anisotropic, the scattered radiation will be depolarized, see, for example, [49]. Of course, it should be kept in mind that the scattering cross section, *σ_sca_*, in this case will be very small, because *σ_sca_* ~ 1/λ^4^ [39] and λ ~ 1 m. Scattered radiation will interact with charged polymer fibers unwound into the bulk of the liquid. We do not know the mass and charge of these fibers, so we do not present formulas that describe the dynamics of these fibers in the field of the incident wave. However, in accordance with the literature data, radiation at frequencies of this range is used in medical practice (in particular, for the treatment of oncological diseases), and this radiation is related not only to electromagnetic waves, but also to acoustic waves; see [50,51]. Based on these indirect data, we can argue that the polymer fibers unwound into the liquid bulk will oscillate at frequency of 100 MHz. In the case of linearly polarized radiation, these oscillations will occur in one plane, but in the case of depolarized radiation, the oscillations will happen in different planes.

As follows from Equation (1), if the luminescence cross section, *σ_lum_*, does not change upon swelling, then the luminescence intensity, *I*(*t*), decreases exponentially. Thus, the stochastic behavior can be explained by random oscillations of *σ_lum_*. To explain this effect on a qualitative level, it is necessary to use the model of non-radiative energy transfer from a donor of luminescence to an acceptor of luminescence; see monograph [52]. Let us imagine that there exists a luminescence center on the membrane surface, which we will call a donor of luminescence. Let us further imagine that at a certain distance, *R*, from the donor there exists another particle (acceptor of luminescence), whose absorption spectrum coincides with the absorption spectrum of the donor. Then, at a certain *R* the process of resonant energy transfer from the donor luminescent level to the acceptor level is possible. An electron from an acceptor level passes into the ground state of an acceptor, which can be accompanied by a photon (luminescence) emission, but a non-radiative transition is also possible. In this case, the luminescence from the donor is quenched. If we consider protein membranes, then donors and acceptors, as a rule, are the same groups of proteins, but in the case of acceptors these groups are slightly changed; see [52]. Because in our case the luminescence centers are sulfonic groups, we can assume that the acceptor is a slightly modified sulfonic group, which is not active with respect to the luminescence. The efficiency, *S*, of energy transfer from a donor to an acceptor is given as:(10)$$S=\frac{R_{0}^{6}}{R_{0}^{6}+R^{6}}.$$

Here, *R*_0_ = 30–60 Å is the so-called Forster parameter; see [52]. Thus, the value of *S* varies as *R*^−6^, i.e., it is a very steep function. Note that if the donor and acceptor are rigidly fixed on the membrane surface, then the distance, *R*, between them is always fixed, i.e., the luminescence cross section, *σ_lum_*, does not change upon swelling. However, in our case, the donor and acceptor are located on the polymer fibers unwound in the liquid bulk, i.e., their spatial position can change due to some external electromagnetic forces (remember that sulfonic groups are charged). If these forces change the distance, *R*, then, according to Equation (10), the value of *S* can be either ~1, and the luminescence stops (the effect of quenching), or *S* << 1, and in this case there is no energy transfer, and the luminescence is quite intense. Thus, the only mechanism in our experiments, due to which the luminescence can disappear/reappear in NaCl solutions based upon natural water, is changing the distance, *R*, which leads to the oscillations of *σ_lum_*. It seems obvious that when the polymer fibers, unwound into the liquid bulk, are driven by linearly polarized radiation, these fibers will vibrate in the single plane. In this case the distance, *R*, between the donor and the acceptor does not change, that is, the value of *σ_lum_* remains constant. At the same time, if these fibers are swayed by depolarized radiation, that is, there exist polarizations in different planes, then the distance, *R*, will change randomly, and stochastic switching of luminescence quenching/excitation modes is possible. Within the framework of this model, the appearance of stochastic phenomena in our experiments is qualitatively explained. The question of why the addition of piperine also leads to the stochastization in the behavior of *I*(*t*) remains unclear.

We can assume that random oscillations of *I*(*t*) can also arise under mechanical disturbances of the unwound polymer fibers. To verify this, the following experiment was performed. We studied doubly distilled water where the bubstons content is very low, and the cluster phase is practically absent; the liquid samples were not processed with electric pulses. A capillary of 0.5 mm thickness was inserted into the cell (not shown in Figure 2), where the Nafion membrane was soaked. After 40 min of soaking, atmospheric air was pumped through this capillary into the cell. This was accompanied by the generation of bubbles that floated up to the surface of water and generated shear capillary waves at the surface. In Figure 14, we show the dependence of *I*(*t*). It is seen that, prior to the pumping of bubbles, the curve, *I*(*t*), coincides to a good accuracy with the reference dependence. As soon as the shear capillary wave was excited, the oscillations of *I*(*t*) started; at a certain moment of time, the luminescence signal dropped to zero. It is very important that, in this experiment, the Nafion membrane was entirely submerged in the bulk of water. As is known, shear sound waves are very strongly attenuated in the liquid bulk; see [53]. Nevertheless, the shear wave generation at the surface was accompanied by the oscillations of *I*(*t*) in the liquid bulk. This wave should be very weak in the bulk, but nonetheless it can change the distance, *R*, between the donor and the acceptor on the unwound polymer fibers. We can argue that Nafion is a highly sensitive polymer matrix: very weak external force, which can change the distance, *R*, and can lead to a noticeable change in the behavior of *I*(*t*). Therefore, depolarized scattered radiation leads to stochastization effects despite a very small scattering cross section, *σ_sca_*.

## 5. Conclusions

The most interesting result of this work is the observation of stochastic regimes of luminescence intensity, *I*(*t*), during swelling of a polymer membrane in aqueous salt solutions subjected to treatment with electric pulses at a certain repetition rate. For the appearance of stochastic regimes, the influence of an external random force is necessary. It was shown that this force has an electromagnetic nature. According to our hypothesis, regular flashes of distant space objects can be the source of this force. Spectral and correlation processing of the time dependences of *I*(*t*) shows that the source of an external electromagnetic wave could be the pulsations of magnetars or white dwarfs, as described in a recent paper [38]. Stochasticization effects are observed only if the polymer fibers are unwound into the bulk of the liquid during swelling of the polymer. A qualitative theoretical model was created, according to which the stochastic behavior of *I*(*t*) arises due to random changes in the luminescence cross section, *σ_lum_*. These changes occur due to the fact that liquid samples contain bubston clusters, which, as a result of treatment with electrical pulses, acquire anisotropic properties. Incident linearly polarized low-frequency radiation, being scattered in liquid samples, becomes depolarized. In this case, the polymer fibers unwound into the bulk of the liquid experience oscillations in the field of an external wave in different planes, which leads to a change in the average distance between these fibers. It is assumed that, in our case, the effects of resonant luminescence energy transfer between the donor and acceptor are possible, and the distance, *R*, between the donor and acceptor, which are localized on unwound fibers, changes randomly. Within the framework of this qualitative model, it is possible to explain the effects of stochastization arising in our experiments.

## Figures and Tables

**Figure 1 polymers-14-00688-f001:**
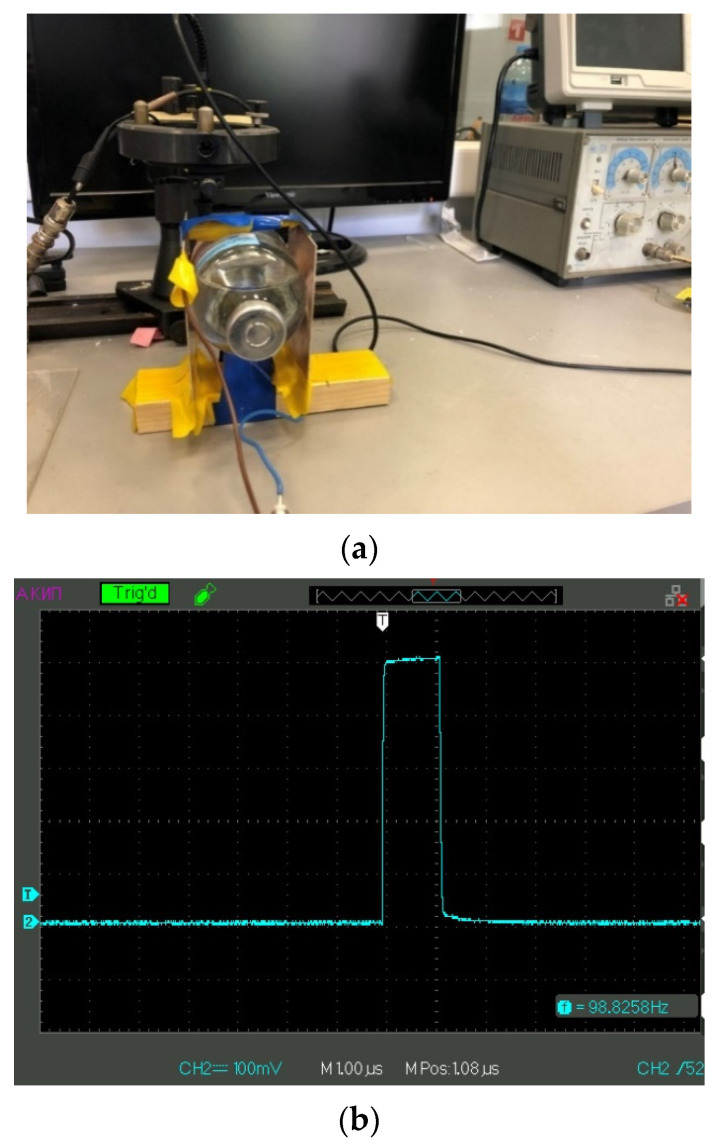
Panel (**a**) irradiation of a test liquid in a flat capacitor. Panel (**b**) the shape of an electric pulse applied to the plates of the capacitor.

**Figure 2 polymers-14-00688-f002:**
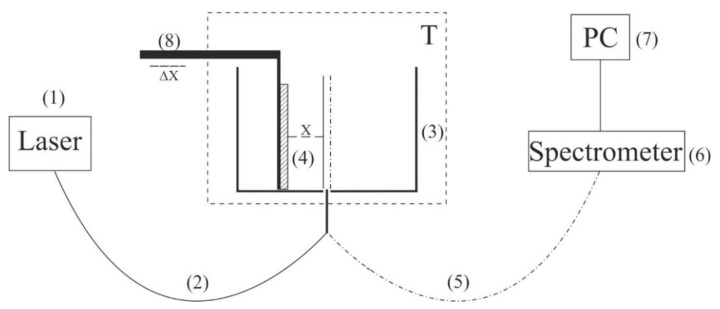
Schematic of the experimental setup for laser luminescence spectroscopy. (1) laser diode; (2) optical fiber; (3) cylindrical cell; (4) Nafion plate; (5) quartz fiber; (6) minispectrometer; (7) computer; (8) stepping motor; T–thermostat.

**Figure 3 polymers-14-00688-f003:**
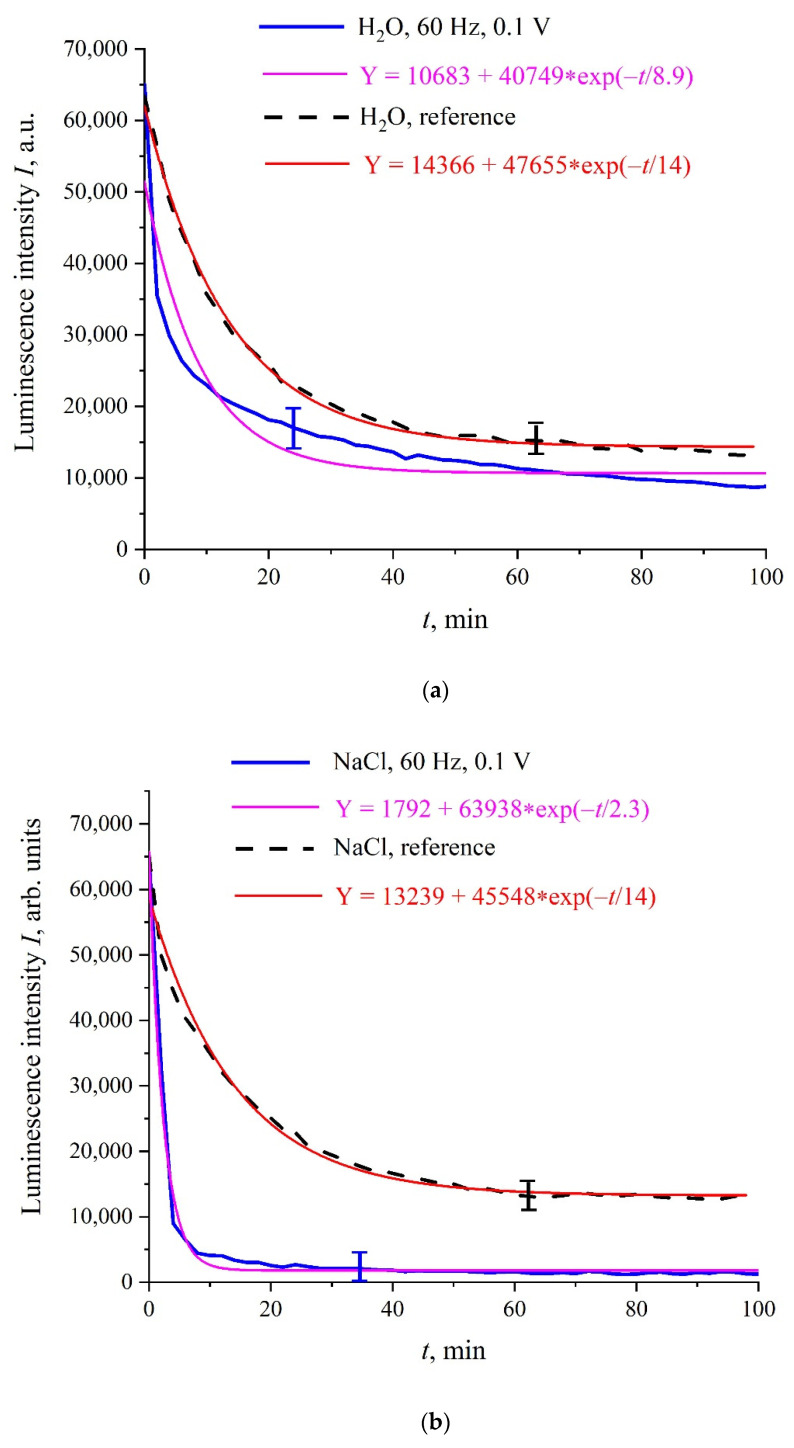
Intensity, *I*, of luminescence in the spectral maximum vs. the polymer membrane soaking time, *t*, for liquid samples, processed with 100 mV amplitude electric pulses at a repetition rate of 60 Hz; unprocessed (reference) samples are highlighted with a dashed line; (**a**)—Milli-Q water; (**b**)—NaCl solution; (**c**)—Ringer’s solution.

**Figure 4 polymers-14-00688-f004:**
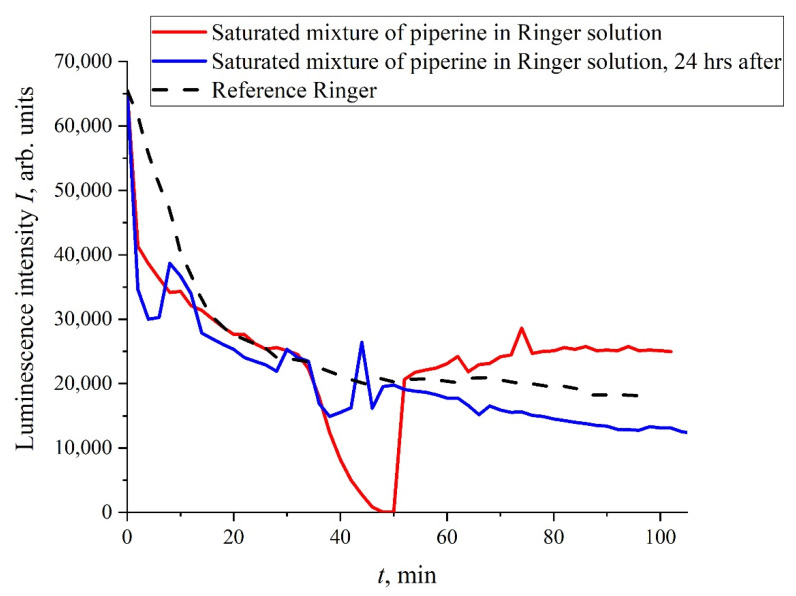
Dependence of *I*(*t*) for isotonic Ringer solution containing piperine with the concentration 40 mg/L (saturated mixture). The red curve is related to the measurements immediately after the piperine diluting, while the blue curve is related to the same measurements, performed 24 hr after.

**Figure 5 polymers-14-00688-f005:**
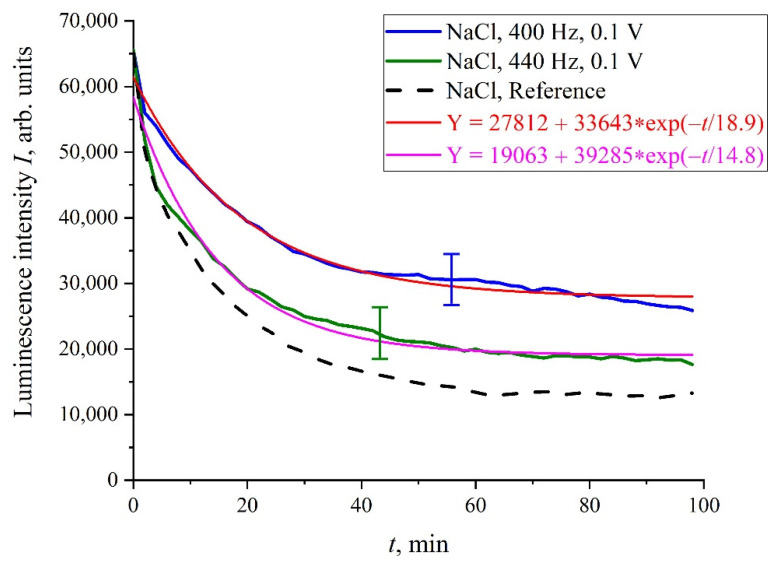
The dependence of *I*(*t*) for NaCl solutions, processed with 100 mV amplitude electric pulses at a 400 and 440 Hz repetition rate.

**Figure 6 polymers-14-00688-f006:**
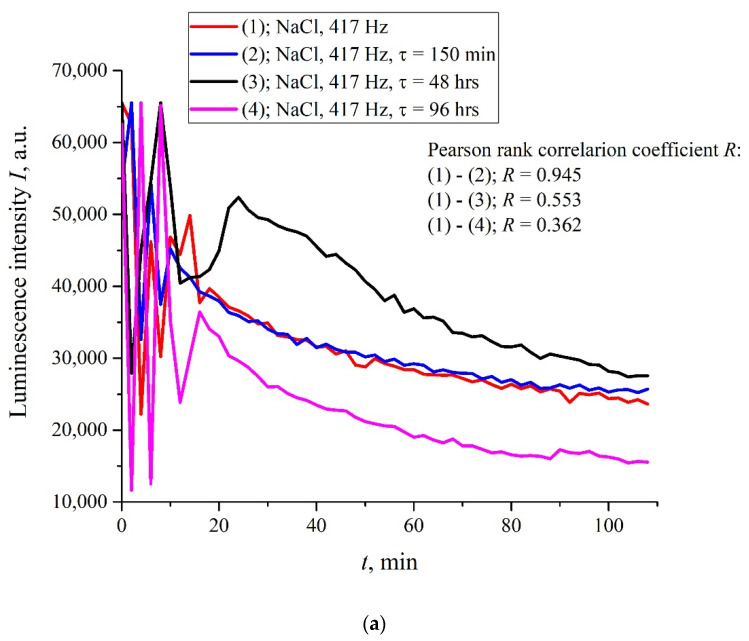
The intensity *I*(*t*) for isotonic NaCl solution after processing at the 417 Hz pulse repetition rate. The difference between the graphs in panels (**a**,**b**) is that the measurements were made in different laboratories; the distance between the laboratories is 20 km. Here, *τ* is the time interval between the beginnings of the corresponding measurements. Pearson rank correlation coefficient was calculated for all values of *τ*.

**Figure 7 polymers-14-00688-f007:**
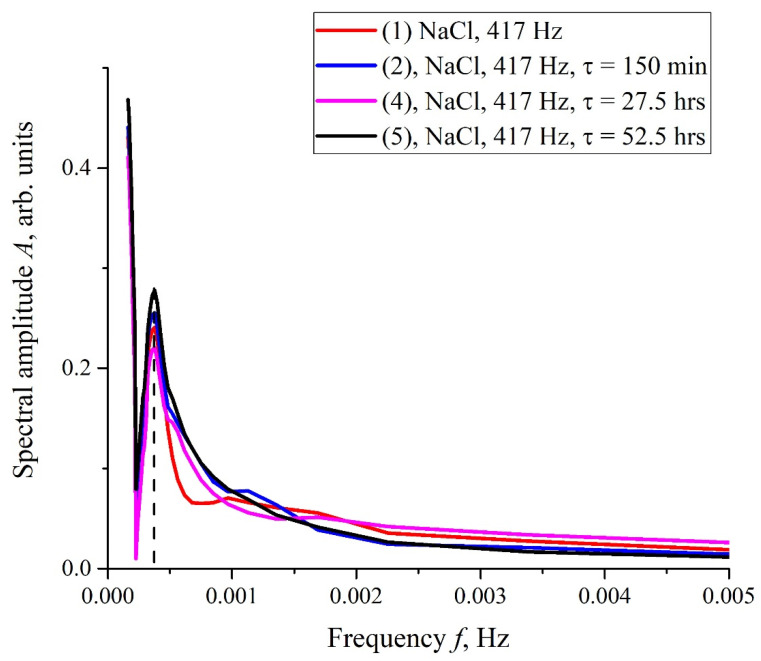
Results of the Fourier transform of *I*(*t*) dependences shown in Figure 6b. The frequency dependences of the spectral amplitude, *A*(*f*), were obtained using the Morlet wavelet transform. The dashed line marks the central frequency of the spectral maximum.

**Figure 8 polymers-14-00688-f008:**
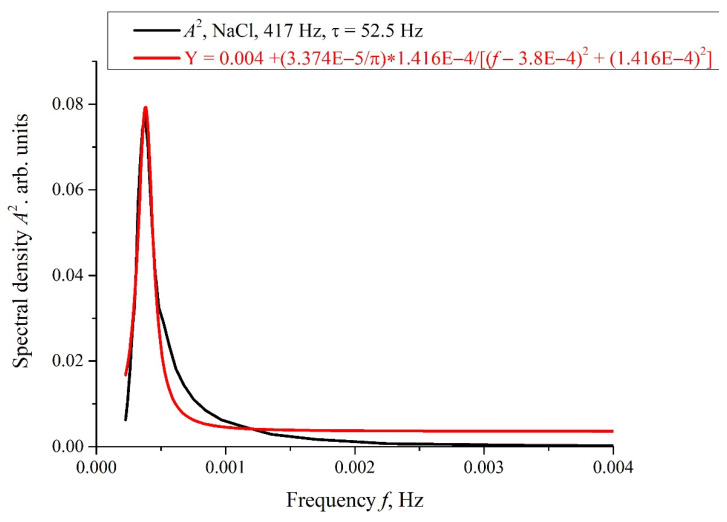
Spectral density, *A*^2^(*f*), for the dependence *A*(*f*) at *τ* = 52.5 h in Figure 7. This dependence is approximated by Lorentzian; see Equation (4).

**Figure 9 polymers-14-00688-f009:**
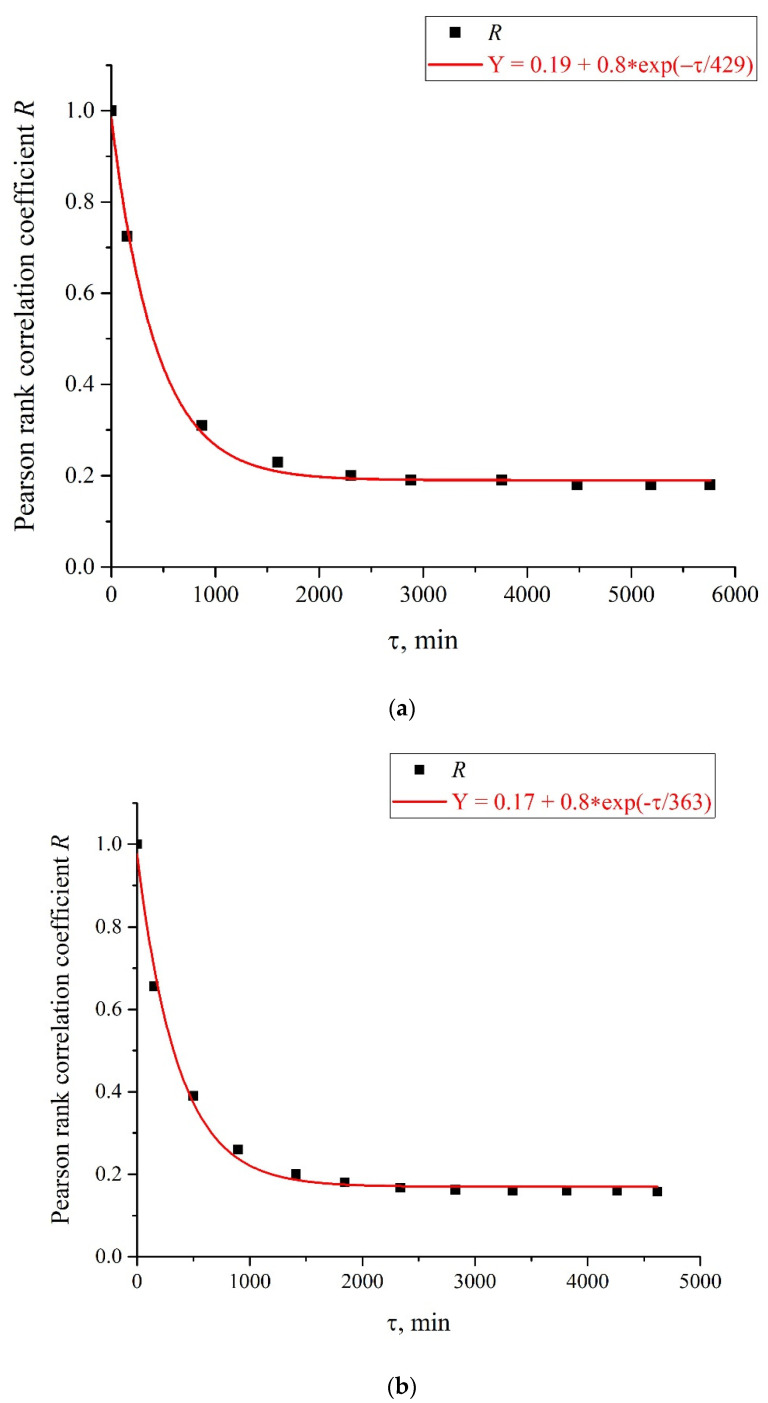
Pearson rank correlation coefficient, *R*(*τ*), for the curves in Figure 6. Panel (**a**) is related to the curves in Figure 6a, and panel (**b**) is related to the curves in Figure 6b.

**Figure 10 polymers-14-00688-f010:**
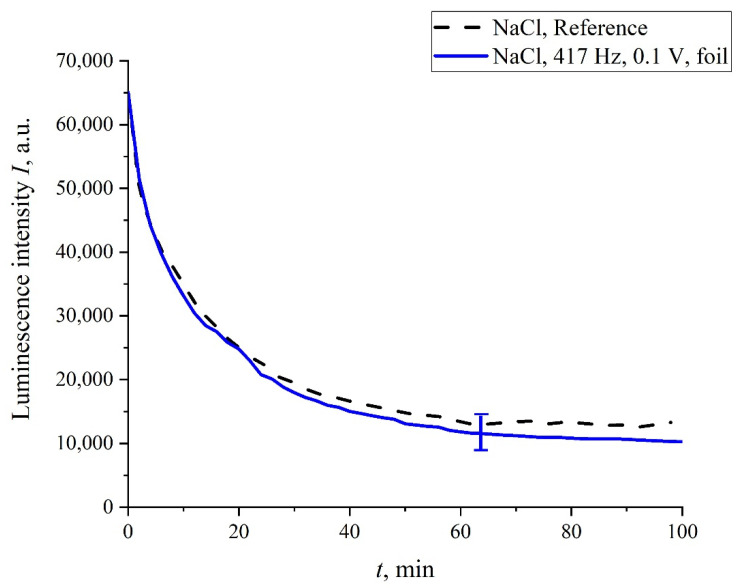
The dependence of *I*(*t*) after processing at a 417 Hz pulse repetition rate for NaCl solution; the liquid sample was wrapped with 10 µm-thick aluminum foil during treatment.

**Figure 11 polymers-14-00688-f011:**
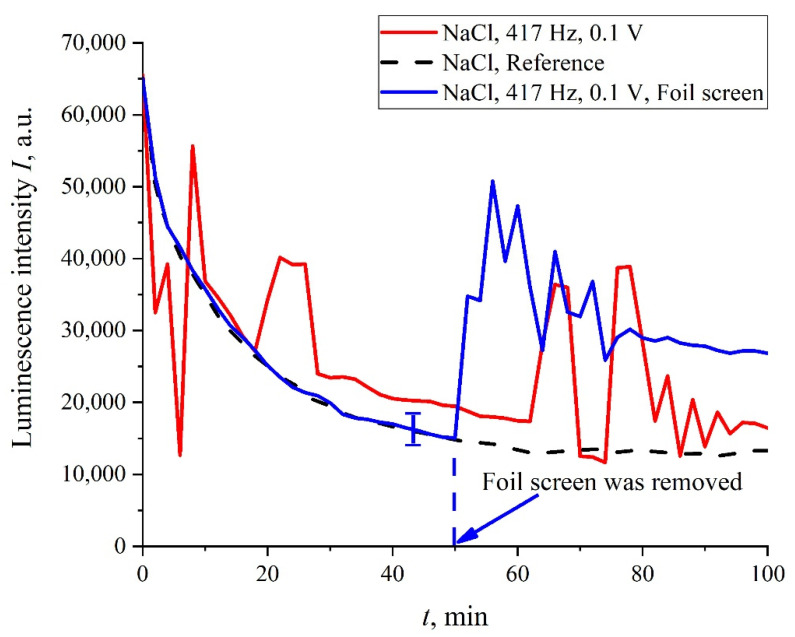
The dependence of *I*(*t*) after processing at a 417 Hz pulse repetition rate for NaCl solution; the photoluminescence setup was covered with a 10 µm-thick aluminum foil screen (blue curve), or non-covered with the screen (red curve). The stochastic behavior is restored immediately after removing the screen.

**Figure 12 polymers-14-00688-f012:**
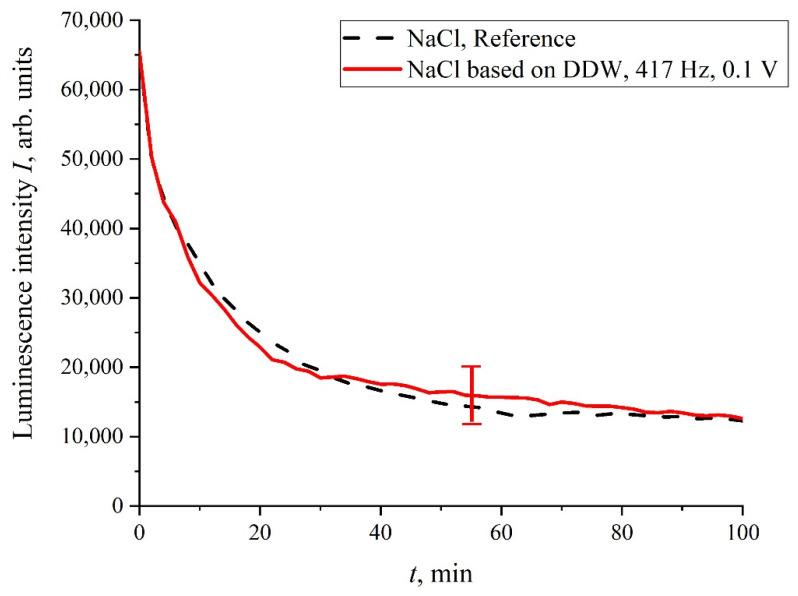
Dependence of *I*(*t*) in DDW-based NaCl solution, processed with the electric pulses at 417 Hz repetition rate.

**Figure 13 polymers-14-00688-f013:**
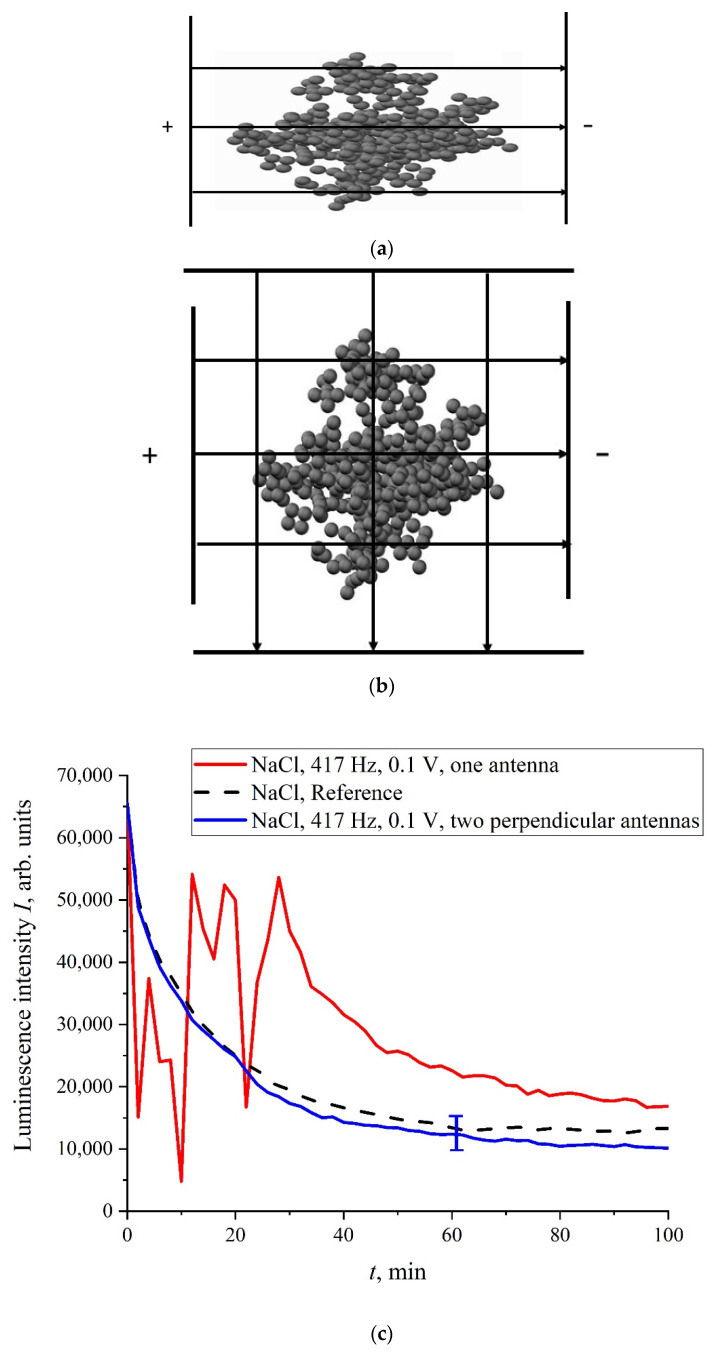
Panel (**a**)—the schematic of processing with one capacitor. Panel (**b**)—the schematic of processing with two capacitors. Panel (**c**)—dependence of *I*(*t*); the NaCl solution was processed beforehand with 417 Hz frequency electric pulses using one/two mutually perpendicular flat capacitors. Red curve is related to one capacitor, whereas blue curve is related to two capacitors.

**Figure 14 polymers-14-00688-f014:**
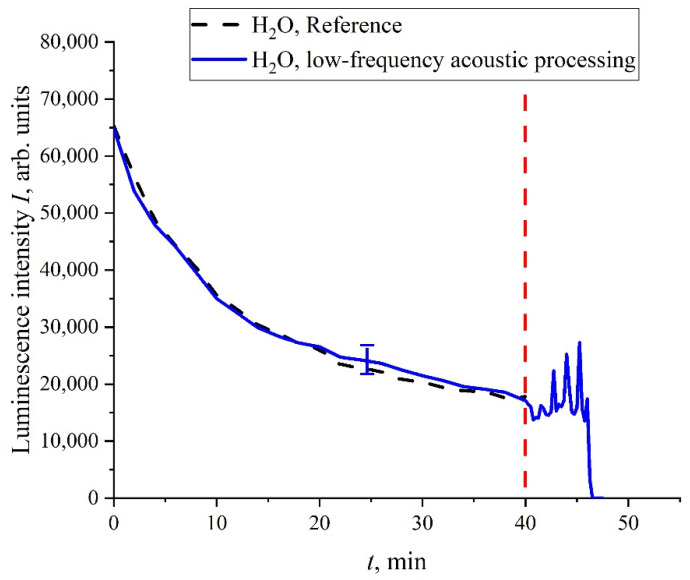
Dependence of *I*(*t*) in water; atmospheric air was pumped after 40 min of soaking.

## Data Availability

The data presented in this study are available on request from the corresponding author.

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
