# Peer review of "Stochastic Ultralow-Frequency Oscillations of the Luminescence Intensity from the Surface of a Polymer Membrane Swelling in Aqueous Salt Solutions"

_polymers, 2022, doi:10.3390/polym14040688_

Round 1

Reviewer 1 Report

Manuscript ID: polymers-1550038

Title: Stochastic ultralow-frequency oscillations of the luminescence

intensity from the surface of a polymer membrane swelling in aqueous salt

solutions.

Authors: Nikolai F. Bunkin *, Polina N. Bolotskova, Elena V. Bondarchuk,

Valery G. Gryaznov, Valeriy A. Kozlov, Oleg V. Ovchinnikov, Oleg P. Smoliy,

Igor F. Turkanov, Catherine A. Galkina, Alexandr S. Dmiriev, Alexandr F.

Seliverstov

Submitted to section: Polymer Membranes and Films, 

Dear Authors,

 Thank you for the opportunity to read your article. I found the topic is interesting and fundamental. Generally speaking, there are some results presented in order to capture some trends but they can be better described and discussed. The conclusion and results need more clear explanation and/or more precision. I suggest that this article will be revised before re-submission. I recommend its major revision at this state. My comments are below. I hope they will be helpful.

Good luck

 “Keywords”

-Please consider listing keywords that are not used in the article title.

 “1. Introduction”

In general, literature of other authors is very limited. A lot of own works are cited, it is not very professional. Please consider citing more literature.

Please consider shaping and reducing the volume of the introduction directly relevant to the contents of the work presented in this article.  

Please consider revising manuscript with providing more details about the originality of your work.

Please consider clearly defining your work goals in terms of their applicability. 

“2. Materials and Methods”

Give conductivity od double distilled water

Double distilled water in abstract and MilliQ in the Materials section means the same, please precise, it is very important for potential readers?

Regarding your zeta potential measurement, please consider some details of methodology, introducing particle concentration, type and concentration of background electrolyte, etc Investigation of the electrokinetic properties of paraffin suspension. 1. In inorganic                    electrolyte solutions ,  Langmuir 21 (2005) 10, 4347-4355

Give some examples of similar studies of other authors.

 A.V. Delgado (Ed.), Interfacial Electrokinetics and Electrophoresis”, Marcel Dekker, New York 2001 Ch.32, pp. 893-931

“The electrokinetic  and rheological behaviour of phosphatidylcholine-treated TiO2 suspensions” Colloids Surfaces A: 440 (2014) 110-115

“3. Results”->3. Results and discussion

In general, the results should be describe and discussed with more details and fair point of view. Please consider revising this section significantly. Please see my detail comments below.

 Fig. 1 make an overlay for this photo with English translations because it is Cyrillic or trim them appropriately.

Fig. 7 There are too few units on the X axis

Equations

I couldn't review the equations because I had them scattered or overlapped, they need to be corrected

Compare with biological membranes ….Please add some fragments about  membrane properties e.g.

What affects  the   biocompatibility of biomaterials? Advances in Colloid and Interface Science 2021, 294, 102451

“4. Conclusions”.

 Precise which isotonic liquids, precise which test liquids, conclusion is too general. Remember some readers read only conclusion, must encourage the reading of the entire work. This phenomenon is unclear but …. we have an idea how to verify it

or probably the mechanism is ... You can't give up like that right now

Literature part should be corrected (now it looks like authoplagiarism)

Minor concerns:

Please polish English more.

Final conclusion:

Major revision

Author Response

We are grateful to the reviewer for a careful reading of the manuscript and their comments. As a result of the criticism expressed, the manuscript was completely rewritten. The main innovation of the latest version of the manuscript: it was possible to connect the stochastic regimes of luminescence intensity observed by us with an external source: the magnetar pulsation described in the work of Hurley-Walker, N.; Zhang; X.; Bahramian, A.; McSweeney, S.J.; O'Doherty, T.N.; Hancock, P. J.; Morgan, J. S.; Anderson, G.E.; Heald, G. H.; Galvin, T.J. A radio transient with unusually slow periodic emission. Nature 2022, 601, 526 – 530 (reference [38]). Unfortunately, this work was not known to us when writing the first version of the manuscript, since it was published on 01/26/2022. Taking into account the new theoretical model, the Discussion and Conclusion sections were completely rewritten.

Below we provide a list of particular corrections that the reviewer suggested. The referee’s comment are highlighted with Italic font.

“Keywords”

Please consider listing keywords that are not used in the article title. This was corrected.

 “1. Introduction”

In general, literature of other authors is very limited. A lot of own works are cited, it is not very professional. Please consider citing more literature.

We have left only 5 references to our works. At the same time, the list of cited literature has increased and contains 53 references in the new version.

Please consider shaping and reducing the volume of the introduction directly relevant to the contents of the work presented in this article. 

The volume has been reduced and rewritten in accordance with the recommendations of the reviewer.

Please consider revising manuscript with providing more details about the originality of your work.

The novelty lies in the detection of stochastic regimes of luminescence intensity during swelling of a polymer membrane in liquid samples treated at certain electrical pulse repetition frequencies. In addition, we investigate liquid samples with different isotopic compositions. This allowed us to detect a number of isolated frequencies of electromagnetic processing, at which the luminescence intensity, usually described by a regular function of time, exhibits a sporadic temporal behavior. Besides, our work consists the study of spectral characteristics of the stochastic dynamics of luminescence and the qualitative theoretical model underlying the stochastic regimes, explaining the differences in swelling kinetics of polymer in natural water and DDW. This is specifically noted in the new version of the Introduction section.

Please consider clearly defining your work goals in terms of their applicability. 

An appropriate comment has been inserted in the Introduction section. It was noted that the results obtained can be used to clarify the mechanisms of interaction of the polymer membrane with isotonic solutions with different isotopic compositions in the field of external low-frequency irradiation, which can be applied in medical practice.

“2. Materials and Methods”

Give conductivity of double distilled water.

The Nafion plates were soaked in Milli-Q water with a resistivity of 4 MW´cm (measurement were made 1 hr after the preparation). Conductivity measurements were made using a Eutech CON270043S conductometer, Thermo Fisher Scientific, Waltham, MA, USA.

Double distilled water in abstract and MilliQ in the Materials section means the same, please precise, it is very important for potential readers?

Sorry for that. It is obviously the same. This misunderstanding was fixed.

Regarding your zeta potential measurement, please consider some details of methodology, introducing particle concentration, type and concentration of background electrolyte, etc Investigation of the electrokinetic properties of paraffin suspension. Give some examples of similar studies of other authors.

We do not describe experiments on measuring the zeta potential in this section. We have, however, referred to these works in the section where we refer to experiments on the measurement of zeta potentials. Below is an excerpt from the list of cited literature.

  1. Dustan, D.E. The Electroforetic Mobility of Hydrocarbon Particles in KCl Solutions, Langmuir 2005, 21, 4347-4355.
  2. Delgado A.V. Interfacial electrokinetics and electrophoresis; Ch. 32; Marcel Dekker, New York 2002.
  3. Wiącek, A.E.; Anitowska, E.; Delgado, A.V.; Hołysz, L.; Chibowski, E. The electrokinetic and rheological behavior of phosphatidylcholine-treated TiO2 suspensions, Colloids Surfaces A 2014, 440, 110-115.
  4. Results and discussion

In general, the results should be described and discussed with more details and fair point of view. Please consider revising this section significantly. Please see my detail comments below.

 Fig. 1 make an overlay for this photo with English translations because it is Cyrillic or trim them appropriately.

This was corrected.

Fig. 7 There are too few units on the X axis.

This was fixed.

Equations

I couldn't review the equations because I had them scattered or overlapped, they need to be corrected.

The equations were corrected in part. To avoid repetitions, we changed the notation in some formulas. However, the formulas themselves in this article were not derived by us. These formulas are either well known or have been derived in other papers to which we refer.

Compare with biological membranes. Please add some fragments about membrane properties e.g. What affects the biocompatibility of biomaterials? Advances in Colloid and Interface Science 2021, 294, 102451

We are grateful to the referee for pointing to this work. Unfortunately, this review was not known to us. We refer to this review in the Introduction, ref. [20].

“4. Conclusions”.

Precise which isotonic liquids, precise which test liquids, conclusion is too general. Remember some readers read only conclusion, must encourage the reading of the entire work. This phenomenon is unclear but …. we have an idea how to verify it

or probably the mechanism is ... You can't give up like that right now

Literature part should be corrected (now it looks like authoplagiarism)

The Conclusion section was completely rewritten.

Minor concerns:

Please polish English more.

The quality of English presentation was improved.

Reviewer 2 Report

Authors studied the photoluminescence from the surface of the Nafion polymer membrane during swelling in isotonic solutions and doubly distilled water. This paper is a continuation of the previous article published in the Polymers Journal in the 2021 year. Authors should made some changes and additions in the text.

Keywords

  • The keyword "bubstons" is not necessary.

Introduction

  • The state of the art has been correctly presented.

Materials and Methods

  • Description on the Fig. 1 in the English language should be given.
  • The formula in the 2.2.2. section is not clear.

Experimental Results

  • Formulas are with very poor quality. Authors should change fonts and add explanation directly after each formula.
  • All curves with the formulas and R2 coefficient should be described.
  • This section is too long. Authors should consider the division into shorter parts.

Discussions

  • Authors compared results mainly with their previous investigations. Is the topic only popular in one country?

Conclusions

  • Conclusions are short. Authors should consider longer summary.

References

  • Authors cited too much their own articles. They should change the relationship between own/total amount of papers.

I recommend the paper for the publishing after minor changes and additions.

Author Response

We are grateful to the reviewer for a careful reading of the manuscript and their comments. As a result of the criticism expressed, the manuscript was completely rewritten. The main innovation of the latest version of the manuscript: it was possible to connect the stochastic regimes of luminescence intensity observed by us with an external source: the magnetar pulsation described in the work of Hurley-Walker, N.; Zhang; X.; Bahramian, A.; McSweeney, S.J.; O'Doherty, T.N.; Hancock, P. J.; Morgan, J. S.; Anderson, G.E.; Heald, G. H.; Galvin, T.J. A radio transient with unusually slow periodic emission. Nature 2022, 601, 526 – 530 (reference [38]). Unfortunately, this work was not known to us when writing the first version of the manuscript, since it was published on 01/26/2022. Taking into account the new theoretical model, the Discussion and Conclusion sections were completely rewritten.

Below we provide a list of particular corrections that the reviewer suggested. The referee’s comment are highlighted with Italic font.

Keywords

The keyword "bubstons" is not necessary.

This keyword was removed.

Materials and Methods

Description on the Fig. 1 in the English language should be given.

This annoying mistake was fixed.

The formula in the 2.2.2. section is not clear.

It is noted in this formula that the luminescence intensity is proportional to the number of luminescence centers, which must be multiplied by the luminescence cross section and by the pump intensity. The number of luminescence centers is the volumetric density of these centers multiplied by the volume occupied by the pump radiation. Since our experiments are done in grazing incidence geometry, this volume is equal to the cross section of the pump beam, which must be multiplied by the height of the Nafion plate. This formula also includes the dimensional constants A and k, which are not known to us, since the results are presented in relative units. We did not include such a detailed description in the new version of the article, since this formula was deduced in our previous article [16].

Experimental Results

Formulas are with very poor quality. Authors should change fonts and add explanation directly after each formula.

When typing formulas, we used the MathType software. However, if the manuscript is accepted for printing, all formulas will need to be retyped in the MSWord formula software. Therefore, this error will be corrected if the manuscript is accepted for publication.

All curves with the formulas and R2 coefficient should be described.

This was done in the new issue.

This section is too long. Authors should consider the division into shorter parts.

The section was reduced.

Discussions

Authors compared results mainly with their previous investigations. Is the topic only popular in one country?

We have left only 5 references to our works. At the same time, the list of cited literature has increased and contains 53 references in the new version.

Conclusions Conclusions are short. Authors should consider longer summary.

The Conclusion section was completely rewritten.

References

Authors cited too much their own articles. They should change the relationship between own/total amount of papers.

We have left only 5 references to our works. At the same time, the list of cited literature has increased and contains 53 references in the new version.

Reviewer 3 Report

The manuscript includes the random luminescence from Nafion membrane. The content has partially interesting topic, especially 'exclusion zone' from polymer-brush-like structure, but had to understand them. The story and something you want to propose is not clear. Unfortunately, the manuscript is at rejected. The comments are as follows,

1) in abstract, i can not understand the meaning. line 21-25, this occurs on or in the membrane?line 30, what is the 'random function'? line 30, what is 'unwound in the bulk....'? line 34, where is the distance? where is acceptor and donor(in membrane?). Anyway, please reconstruct the abstract.

2) in introduction, firstly you want to show 1 exclusion zone, 2 size of polymer -brush domain, 3 size of EZ from Nafion membrane, and 4 suddenly mention the luminescence quenching.... I am not sure why quenching takes place? where and how?

3) I think journal is a kind of chemistry, and is different from physics. Thus, in introduction, please illustrate the image you want to find, or propose.

Author Response

We are grateful to the reviewer for a careful reading of the manuscript and their comments. As a result of the criticism expressed, the manuscript was completely rewritten. The main innovation of the latest version of the manuscript: it was possible to connect the stochastic regimes of luminescence intensity observed by us with an external source: the magnetar pulsation described in the work of Hurley-Walker, N.; Zhang; X.; Bahramian, A.; McSweeney, S.J.; O'Doherty, T.N.; Hancock, P. J.; Morgan, J. S.; Anderson, G.E.; Heald, G. H.; Galvin, T.J. A radio transient with unusually slow periodic emission. Nature 2022, 601, 526 – 530 (reference [38]). Unfortunately, this work was not known to us when writing the first version of the manuscript, since it was published on 01/26/2022. Taking into account the new theoretical model, the Discussion and Conclusion sections were completely rewritten.

Below we provide a list of particular corrections that the reviewer suggested. The referee’s comment are highlighted with Italic font.

in abstract, I cannot understand the meaning. line 21-25, this occurs on or in the membrane? line 30, what is the 'random function'? line 30, what is 'unwound in the bulk....'? line 34, where is the distance? where is acceptor and donor (in membrane?). Anyway, please reconstruct the abstract.

The abstract has been rewritten and any mentions of luminescence donors and acceptors have been removed.

in introduction, firstly you want to show 1 exclusion zone, 2 size of polymer -brush domain, 3 size of EZ from Nafion membrane, and 4 suddenly mention the luminescence quenching.... I am not sure why quenching takes place? where and how?

Any mentions of luminescence quenching have been completely removed from the Introduction section. Apparently, we did not quite clearly state in the Introduction the relationship between the excluded zone, the unwinding of polymer fibers and the structure of the stiff brush type. In the new version of the manuscript, in the Introduction section, we write that, in accordance with our model, the Exclusion Zone effect (the expulsion of colloidal microspheres from the Nafion surface into the water bulk) is due precisely to the unwinding of polymer fibers and the formation of a stiff brush-type structure.

I think journal is a kind of chemistry, and is different from physics. Thus, in introduction, please illustrate the image you want to find, or propose.

The Introduction has been completely rewritten to accommodate the criticism. Our work relates rather to physical chemistry, that is, to the science of materials (more specifically, polymers) and the interaction of polymers with low-frequency radiation.

Round 2

Reviewer 1 Report

Good job!

Reviewer 3 Report

The manuscript is revised against the comments.